# VIOLA jones algorithm with capsule graph network for deepfake detection

Venkatachalam K[1], Pavel Trojovský[2] and Štěpán Hubálovský[1]

[1] Department of Applied Cybernetics, Faculty of Science, University of Hradec Králová, Hradec Králová, Czech Republic
[2] Department of Mathematics, Faculty of Science, University of Hradec Kralove, Hradec Kralove, Czech Republic

## ABSTRACT

DeepFake is a forged image or video created using deep learning techniques. The present fake content of the detection technique can detect trivial images such as barefaced fake faces. Moreover, the capability of current methods to detect fake faces is minimal. Many recent types of research have made the fake detection algorithm from rule-based to machine-learning models. However, the emergence of deep learning technology with intelligent improvement motivates this specified research to use deep learning techniques. Thus, it is proposed to have VIOLA Jones's (VJ) algorithm for selecting the best features with Capsule Graph Neural Network (CN). The graph neural network is improved by capsule-based node feature extraction to improve the results of the graph neural network. The experiment is evaluated with CelebDF-FaceForencics++ (c23) datasets, which combines FaceForencies++ (c23) and Celeb-DF. In the end, it is proved that the accuracy of the proposed model has achieved 94.

## INTRODUCTION

Deep fake-based face detection is a process of analyzing the face that is fake or real using deep learning algorithms. Due to the advantages and high effectiveness, this research has used a VIOLA Jones based Capsule graph neural network. VIOLA Jones's approach is a novel technique to detect objects rapidly offline and in real time. It can process the data in 15 frames per second. It works by calculating Gaar features, ada boost processing, and cascaded filters. A capsule graph neural network is a novel deep-learning technique inspired by node embeddings. Moreover, node features are extracted as a capsule and graph that is used to fetch the necessary information.

The Internet world requires advanced fake face detection in several applications, such as social media, forensics, and face-related security applications. Most face detection research suggests using a convolutional neural network (CNN) for recognizing objects. AlexNet (*Krizhevsky, Sutskever & Hinton, 2012*), ResNet (*He et al., 2016*), and Xception (*Chollet, 2017*) are advanced object detection models used in research papers. This model is successful due to its stacked layer in the architecture. Low-level features are extracted, and object-specific features are classified (*Zhang et al., 2021a*). In fake face

Corresponding author
Venkatachalam K,
venkatme83@gmail.com

classification, transfer learning is widely used in detecting faces. However, the Xception technique is as powerful as fake detection methods (*Rössler et al., 2018*; *Khodabakhsh et al., 2018*).

Technological advancement has made forging the images and videos as real ones which look like original frames. Deep and machine learning models are broadly used in faking images, content, and videos. The technology of faking celebrities' essential pictures and videos makes society's most significant issue. Pornographic pictures and images of politicians are modified to spread a wrong impression about the targeted person in the community (*Rössler et al., 2019*; *Zhou et al., 2019*; *Cheng et al., 2022*). Sometimes fraudsters misuse the images with sensitive issues. This is considered a social problem. At present, there is a need for effective technology to predict simple modifications in the images (*Zhuang et al., 2022*; *Zhuang, Jiang & Xu, 2022*).

The publication posts (*Wang & Deng, 2021*; *Zhou et al., 2021*; *Liu, Zhang & Lu, 2020*) of Washington DeepFakes are shared among the public. Some face images are very skillfully modified and circulated on social media without the knowledge of the owner of the image. Additionally, more software and companies are available for performing deep fake activities (*Wang & Deng, 2021*; *Liu et al., 2021a*). Thus, it is necessary to pay attention to research techniques to detect deep fake faces for security and integrity. In this regard, deep learning neural network techniques are widely suggested methods for identifying fake images (*Wang et al., 2022a*; *Dong et al., 2021*; *Lin et al., 2022*). The primary concern is that the training time is prolonged, but the detection time is accurate and fast. To overcome issues and improve the detection of fake faces, this article proposes the VIOLA Jones technique with a capsule dual graph neural network model.

The capsule graph neural network overcomes the embedding problem by addressing nodes and tracing the graph in a neural network. With the advantages of VIOLA Jones with the capsule graph neural network, this research contributes in the following ways:

1. We are improving detection accuracy using intelligent tracing and graph models for fake detection.
2. Node features are detected and trained fast so that the computation time of the proposed work is reduced.
3. The routing mechanism in node features improves the accuracy of the detection.

Further, this research article is presented in five sections. Section 2 has literature work on deep fake detection techniques. Section 3 has an implementation of algorithms and discussions. Section 4 has the experimental outcome. Section 5 ends with the conclusion and future scope of the research.

## LITERATURE REVIEW

Deep fake detection requires face recognition to be critical (*Wang & Deng, 2021*; *Lin et al., 2022*; *Li et al., 2017*) in identifying fake frames and manipulations. Some gestures such as happiness, anger, fear, sadness, disgust, and neutrality are recognized in faces. Human face recognition using data augmentation and convolutional neural networks is fine-tuned in the article (*Umer et al., 2022*; *Hossain et al., 2021*; *Zhang et al., 2021b*). Another side,

biometric-based anti-spoofing techniques in face recognition are surveyed (*Galbally, Marcel & Fierrez, 2014*; *Zheng et al., 2022*; *Mi et al., 2022*). The combination of data preprocessing, image feature selection, and classification techniques are employed in fake detection (*Umer, Dhara & Chanda, 2019*; *Zong, Wang et al., 2022*; *Zhao et al., 2022*). Face image landmark is retrieved from faces for notifying the person. Finally, features detected regions are used to extract the parts and scores calculated for the facial area (*Xu et al., 2022a*; *Huang et al., 2022*; *Zhou & Zhang, 2022*). In real-time detection, the above techniques take much time. Also, the accuracy of detection is still a research question. VIOLA Jones's algorithm can perform face detection in real-time with high accuracy. Also, the accuracy of detection is still a research question. The VIOLA-Jones algorithm can perform face detection even in real time with high accuracy.

Feature selection using PRNU for extracting face images are traditional techniques (*Lugstein et al., 2021*; *Rathgeb et al., 2020*; *Scherhag et al., 2019*). The SVM classification is used to classify the fake and real from the dataset. Mesoscopic techniques for facial forgery detection using two methods, meso-4 and misconception, are suggested in the article (*Afchar et al., 2018*; *Zheng, Liu & Yin, 2021*; *Tian et al., 2021a*). Besides, it can detect hyper-realistic forged images and videos on DeepFake. The deep neural network for deepfake detection in existing models can be classified into three types. A fine-tuning technique is employed so as to improve generalization models. Combination of LSTM-CNN (*Güera & Delp, 2018*; *Zong & Wan, 2022*; *Tian et al., 2021b*) with InceptionV3 used to detect face swaps using frame level features. Features are processed in CNN, and its output is fed as input to LSTM. Finally, detection accuracy is achieved at 97%.

Deep learning-based capsule network (*Nguyen, Yamagishi & Echizen, 2019*; *Chang et al., 2022*; *Chen, Du & Guo, 2021*) is used for detecting forged images, videos, forged paintings/ scenarios, detection of reply attacks, and computer-designed pictures, audio, and videos. The computer vision challenges are addressed using capsule technology in forensic problems. Dynamic routing technique with capsule network (*Sabour, Frosst & Hinton, 2017*; *Li, Du & Wei, 2021*; *Xie et al., 2022*) is being used to represent relationship hierarchal with demonstrating object pieces. Three capsules are used in routing the images and classifying the real and fake models (*Ma et al., 2021*; *Wang et al., 2017*).

Artifacts with generative adversarial networks (GAN) force the forensic classifier (*Xuan et al., 2019*; *Wang et al., 2022b*) to detect GAN images. The preprocessing step uses gaussian noise and blur methods to remove the unwanted features of PGGAN PGGAN (*Creswell & Bharath, 2018*), DCGAN (*Radford, Metz & Chintala, 2015*). WGAN-GP (*Karras, Laine & Aila, 2019*) are highly used to generate GAN pictures from the CelebA-HQ. Then the generated images are trained in PGGAN and other above methods. Three datasets Face2Face, FaceSwap, and DeepFake in *Dang et al. (2020)* and *Xu et al. (2022b)* use a recurrent neural network with CNN and detect the manipulated frame very fast. The CNN with ResNet model (*Neves et al., 2020*; *Choi et al., 2018*), DenseNet (*He et al., 2016*; *Sabir et al., 2019*; *Chen et al., 2020*) and bidirectional recurrent technology is utilized to achieve higher accuracy. A survey on deepFake technologies is shown in Table 1.

The Gabor filters are widely used in sensitive applications like remote sensing, medical diagnosis, threat applications, *etc*. To improve the efficiency of the Gabor technique,

**Table 1 Survey on deepfake technologies.**

| Article | Methods | Type of data | Technology | Dataset |
|---|---|---|---|---|
| Hu, Li & Lyu (2021) | Physical constraints | Fake Image | Hough transform, canny edge detection | FFHQ |
| Demir & Ciftci (2021) | Eyes consistency | Fake videos | Dense 3 other network | FF++,Celeb-DF |
| Nirkin et al. (2021) | Two region analysis | Fake image | Xception | FF++,Celeb-DF |
| Chugh et al. (2020) | Modality analysis | Fake image | MDS networks | DFDC,DF-TIMIT |
| Chai et al. (2020) | Patch based concept | Fake image | ResNet,Xception, CNN | CelebA-HQ, |

the Fusion technique is applied to the Gabor filter for segmenting the remote sensing images (*Liu et al., 2021b*; *Zheng et al., 2021b*; *Shi et al., 2022*). Also, the convolutional neural network is used with the Gabor filter for detecting the deepFake images on social media.

# PROPOSED VIOLA-JONES BASED CAPSULE NEURAL NETWORK METHODOLOGY

Detection of a deepfake image is a process of identifying the forged image or video using the VIOLA-Jones algorithm with the Capsule Network model (VJ-CN). The overview of this proposed work is given in Fig. 1. Here input dataset is preprocessed and split as testing and training sets. The feature extraction is done using the K-mean algorithm. The fake face detection is classified using VIOLA Jones and Capsule Network.

Figure 1 VIOLA-Jones phases, namely preprocessing, extracting features, and detecting face images using the VIOLA-Jones model with a capsule graph network.

## Pre-Processing

In the detection of a deepfake image accurately, the forged input image undergoes a pre-processing process. The phases involved in pre-processing are given in Fig. 2. Unwanted noises in the frames and pixels are removed in these techniques.

The phases involved in pre-processing include the removal of noise, data augmentation, shear mapping of the image, and rescaling of the image.

### *Removal of noise*

Removing noise in the input forged image accurately detects the fake image. In this work, the Gabor filter is used. A two-dimensional Gabor filter is used, which modulates the Gaussian kernel function by using a sinusoidal plane wave in the spatial domain. The two-dimensional Gabor function is defined as:

$$ga(a,b;\lambda,\theta,\psi,\sigma,\gamma) = \exp\left(-\frac{a'^2+\gamma^2 b'^2}{2\sigma^2}\right)\exp\left(i\left(2\pi\frac{a'}{\lambda}+\psi\right)\right). \tag{1}$$

The real part of the Gabor function is the following:

$$ga(a,b;\lambda,\theta,\psi,\sigma,\gamma) = \exp\left(-\frac{a'^2+\gamma^2 b'^2}{2\sigma^2}\right)\cos\left(2\pi\frac{a'}{\lambda}+\psi\right). \tag{2}$$

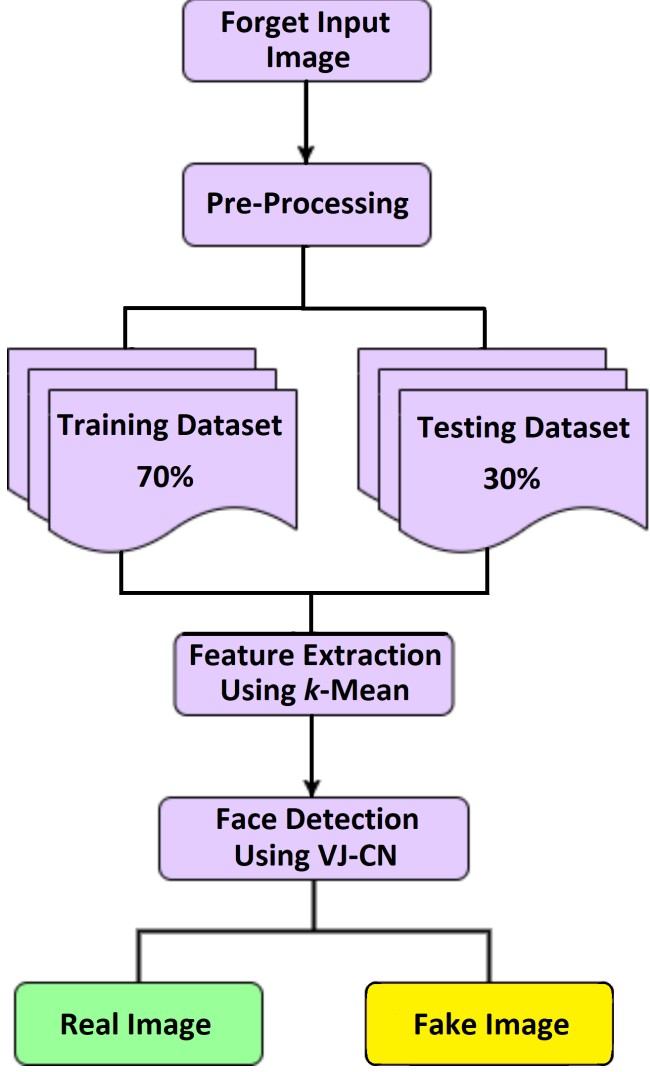

**Figure 1** Overview of VJ-CN.

The imaginary part of the Gabor function is:

$$ga(a,b;\lambda,\theta,\psi,\sigma,\gamma) = \exp\left(-\frac{a'^2 + \gamma^2 b'^2}{2\sigma^2}\right)\sin\left(2\pi\frac{a'}{\lambda} + \psi\right). \tag{3}$$

Here,

$$a' = a\cos\theta + b\sin\theta \tag{4}$$

and

$$b' = -a\sin\theta + b\cos\theta, \tag{5}$$

where $\lambda$ is a sinusoidal wavelength factor. In general, $\lambda$ does not exceed 2, and the orientation of normal to parallel stripes in the Gabor function is defined as $\theta$. The phase

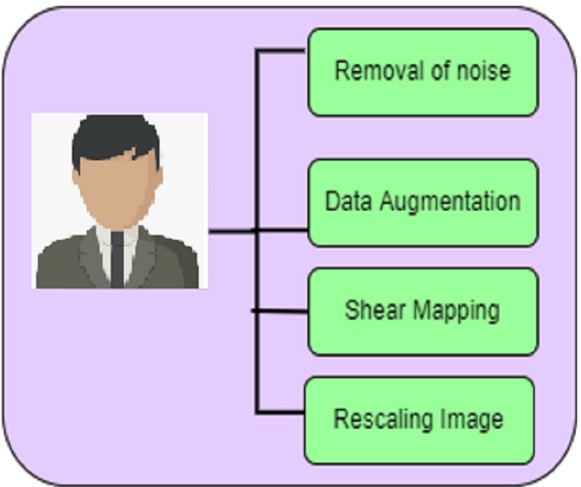

**Figure 2  Preprocessing.**

shift $\psi$ represents the maximum offset in signal modulation, and the aspect ratio $\gamma$ specifies the ellipticity of the Gabor function. $\sigma$ is the standard deviation of the Gaussian envelope.

### Data augmentation

This technique is used to increase the amount of data and generate new data points to reduce the overfitting problem. During the training process of the forged input image dataset, the augmentation processes are implemented as follows:

**Zooming augmentation:** The input image is clearly viewed using Zooming augmentation. View or monitor the input image in the zoom effect as larger with the value 0.4 in the range [0.4, 0.8]. Parameter values vary from 1−value to 1+value.

**Horizontal flipping:** Viewing the zoomed image horizontally flipped when its Boolean value is set with 'true',

Random rotation of almost about 30°.

Random contrast, brightness.

Coarse dropout with size 0.02.

### Shear mapping

Each input image is converted to a vertical direction from the edge part. The parametric control values of the angle of deviation of the horizontal line and the displacement rate. The values of the shear range are 0.2.

A shear mapping is a linear transformation that distorts the shape of an object by skewing it along one or more axes. This transformation is accomplished by multiplying the coordinates of each point in the object by a matrix that includes non-zero off-diagonal entries. The amount of shear is determined by the values of these entries, which control the degree of displacement along each axis.

A shear mapping can be represented by a matrix equation. Suppose we have a 2D point with coordinates (x, y). To apply a shear mapping along the *x*-axis by a factor of k, we can use the following matrix equation: $\begin{bmatrix} x' \\ y' \end{bmatrix} = \begin{bmatrix} 1 & 0 \\ k & 1 \end{bmatrix}$.

Here, (x', y') are the coordinates of the transformed point, and the matrix on the right-hand side represents the shear mapping. To apply a shear mapping along the *y*-axis, we would use a similar matrix with k in the (1, 0) position.

Note that these equations assume that the origin of the coordinate system is fixed. If we want to apply a shear mapping with respect to a different point, we would need to first translate the coordinates so that the origin is at that point, apply the shear mapping, and then translate the coordinates back to their original position.

### *Rescaling image*

The input image consists of RGB values from 0 to 255. The values are rescaled to the interval [0, 1] using the 1/255 scaling method to feed them into the proposed model.

## Feature extraction

The extraction of features in face detection means eds advanced analysis algorithm due to its sensitivity. To improve the performance, extracting features such as the eyes, the position of the nose, chin, etc. in this proposed work, the K-means algorithm is implemented. It calculates the distance between two points in the image by the similarity of pixel values. Minimum distance shows that the similarity of the appearance and its pixel values are grouped to form a class of image. After that, each class is classified as a *gr* group of objects into *pix* pixel points in groups. To determine the high similarity of pixel images by evaluating the average of data objects near *pix* pixel points in groups, it is calculated as

$$\mu_p = \frac{\sum_{i=1}^{n}(cl^i - pix)x^i}{\sum_{i=1}^{n}(cl^i - pix)}, \tag{6}$$

where *cl* denotes the group of all nearest points of the point *pix* and $\mu_p$ is called the center point of the point group.

## Face detection (proposed)

For detecting the face image in an effective and fast manner, this proposed technique of VIOLA-Jones with Capsule Network is implemented (VJ-CN). The overview of the proposed work VJ-CN is shown in Fig. 3.

Figure 3 describes that the proposed work contains two phases. In phase 1, it implements the VIOLA-Jones algorithm for detecting face images, and in phase 2, Capsule Graph is implemented to see face images efficiently. During the training phase, it starts with the training dataset with both real faces and forged images. In the training phase, features are extracted from the face image and stored in a file. After the training process of the dataset, the testing phase is implemented. During the testing phase, all the stored features are applied to the forged input image and classified whether the face is real or fake.

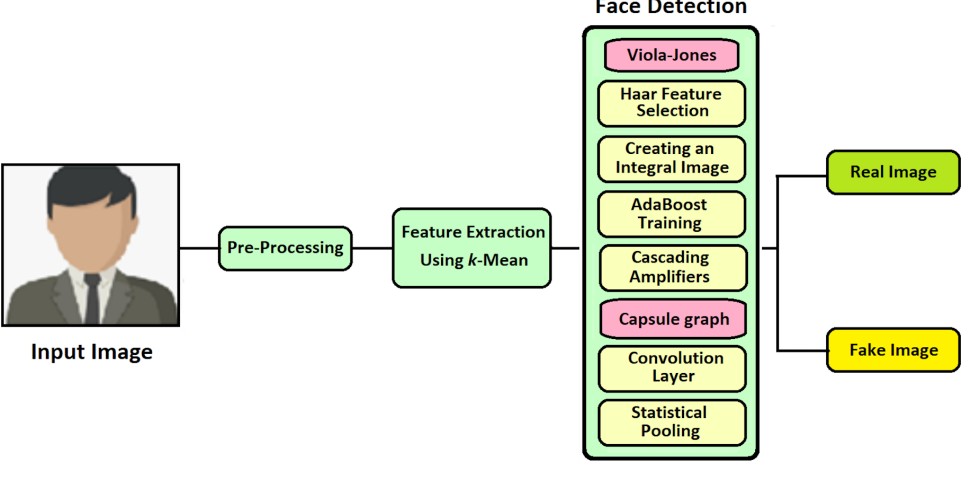

**Figure 3**  Overview of the proposed work.

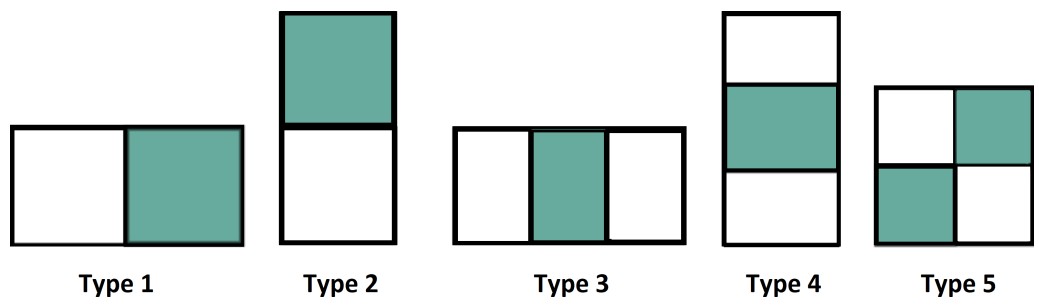

Type 1          Type 2          Type 3          Type 4          Type 5

**Figure 4**  Various types of features.

### The VIOLA-Jones algorithm

A. VIOLA-Jones algorithm is used to detect the face image. This algorithm has four modules; feature selection using the Haar function, integral function, training based on the Adaboost algorithm, and classifier of cascade operator.

**A. Selection of Haar features**

The selection of Haar features is based on the similarity properties of the face image. Some common similarities in face image properties are:

- The size and position of the mouth, eyes, and nose.
- Values of the intensity of pixels.

Figure 4 shows various types of features used in the Haar function.

The intensity value is calculated by the summation of pixel values in the colored area minus the summation of pixel values in the white area and it is defined as.

$$intensity\ value = \sum(Pixels\ in\ colored\ area) - \sum(Pixels\ in\ White\ area). \qquad (7)$$

In VIOLA-Jones, two-rectangular features are used in the detection of the face image. The differences are calculated between colored and white rectangles of the specific area

**Figure 5** Summation area table.

(*Lugstein et al., 2021*; *Zheng et al., 2021a*; *Yu et al., 2019*). The image is to be divided as a sub-window and the features are related on a random basis to the location. Figure 4 shows the extraction of various features from the datasets.

**B. Integral function**

The input image is converted into an integral image by evaluating that every pixel value is equivalent to the cumulative sum of all pixels above and left to that particular input pixel. It can be calculated over input image by sub-window region. The integral of the image is defined below.

$$p(x,y) = img(x,y) + p(x-1,y) + p(x,y-1) - p(x-1,y-1) \tag{8}$$

Figure 5 shows the summation of the particular area table in which Eqn. Eq. (8) is calculated.

Figure 6 shows the sample of the cumulative sum of the particular pixel value of the image.

**C. AdaBoost training**

E. It uses the image's window size and computes the input image's features. The image's most relevant characteristics are selected from the collected features of the image. The features of the image are repeatedly executed for every sub-window size. Thus, it contains a vast set of features. By applying the Adaboost algorithm, it discards the irrelevant features from the featured dataset. These selected features are arranged in weighted order and classified with similar face image pixels. Therefore, these features are called weak classifiers (*Chugh et al., 2020*). In addition to this, the AdaBoost algorithm is implemented based on its threshold value. That is, it is based on the concept of the brute force method through which the trained weak classifier determines the best performer of the feature.

**D. Classifier of the cascade operator**

During the training phase using shear mapping, the cascaded classifier collects the robust classifier by verifying the particular sub-window area that is belonged to the real or the forged image. It is discarded if the sub-window region is classified as a forged image. It is passed on to the next phase in the cascade if it is an actual image. The process is repeated and will be cascaded to get a higher chance of classifying the face image as real or fake. The algorithmic procedure of Viol Jones is given below:

| 1 | 3 | 8 | 10 |
|---|---|---|---|
| 4 | 12 | 7 | 3 |
| 8 | 7 | 3 | 2 |
| 5 | 3 | 1 | 8 |

| 1 | 3 | 8 | 10 |
|---|---|---|---|
| 4 | 12 | 24 | 3 |
| 8 | 7 | 3 | 2 |
| 5 | 3 | 1 | 8 |

**Figure 6** Sample summed area table.

**Algorithm 1: VIOLA-Jones algorithm for the detection of fake or real faces.**
**Input:** Forged image.
**Output:** Face image with similarity indicators.
Step 1: **for** $i = 1$ **to** $n$ **do**
Step 2: Read a down sample $img_{n,n}$.
Step 3: Evaluate the integral image $img_{nn \times nn}$.
Step 4: **for** $j = 1$ **to** No. of sub-windows. **do**
Step 5: **for** $k = 1$ **to** No. of phases in the cascade classifier. **do**
Step 6: **for** $l = 1$ **to** No. of filters of the phase $k$ **do**
Step 7: Create a sub-window and filter the forged image.
Step 8: Accumulate the filtered outputs.
Step 9: **end for** // The variable $l$.
Step 10: **if** Accumulated missed at each phase threshold. **then**
Step 11:    Remove irrelevant features of the face (forged).
Step 12:    Break 'for' loop for the variable $k$.
Step 13: **end if**
Step 14: **end for** // The variable $k$.
Step 15: **if** The sub-window move to the next phase and checks. **then**
Step 16:    Classify the features in the sub-window as Real images.
Step 17: **end if**
Step 18: **end for** // The variable $j$.
Step 19: **end for** // The variable $i$.

The algorithm divides the input forged image into smaller sub-windows. It looks for every feature specifically at all sub-windows and finds the face features. This algorithm checks the image in different positions and scales to identify many faces of different sizes. By implementing this Algorithm 1, the whole image features are scanned using a sub-window, and the face image is detected as real or forged using the capsule graph algorithm. To classify it accurately, the capsule graph network model is implemented. Because of using the V-J algorithm, the face image is detected and for fine-tuning the classification of a real or fake image, it needs to implement a capsule graph network model.

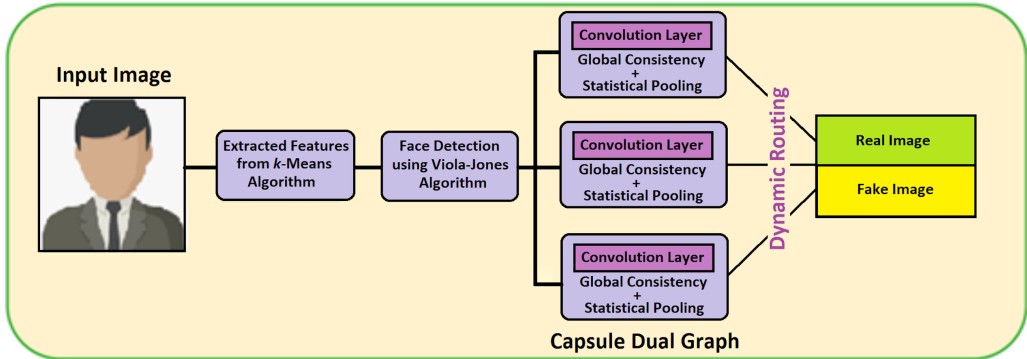

**Figure 7   Capsule graph network.**

### Capsule Graph Network (CGN)

The dual Capsule Graph Network consists of three capsules and for detecting the fake or the real image, two capsules are needed. At first, features are extracted using the k-means algorithm. Furthermore, these features are stored as a file and it endures for further detection in VIOLA-Jones Algorithm. This VIOLA Jones algorithm detects it as a face image and these feature values are fed as input to this CGN model. In the graph, each node represents the features of the face image or video. The dual graph (DGN) network consists of two CNN and the input data is represented as matrix $Id \in \mathbb{R}^{m \times n}$ with the data points $Dp = \{dp_1, dp_2, \ldots, dp_i, \ dp_{i+1}, \ldots dp_n\}$. The structure of the graph is given by the adjacency matrix $A \in \mathbb{R}^{n \times n}$. Vector features and adjacency matrix $A$ are the DGN model inputs. The output of local consistency for hidden layer $i$ is declared in Eq. (6).

$$convo_{LC}^i(P) = z^i = \sigma(P^{-\frac{1}{2}}\overline{A}P^{-\frac{1}{2}}z^{i-1}r^i) \qquad (9)$$

where $\overline{A} = A + I_n$ is the self-loop adjacency matrix of the matrix $A$, $I_n$ is the identity matrix, $P$ is the normalized adjacency matrix, $z$ represents the output of the $(i-1)$th layer, $r$ represents parametric value for the training, and $\sigma$ is the activation function (ReLU). Figure 7 shows the workflow of the Capsule Graph Network.

Figure 7 contains three main capsules and produces two output capsules in terms of real fake images or videos. Features extracted from the k-means algorithm are fed as input to face detection using the VIOLA-Jones algorithm and its output is passed as input to the DGN model. Therefore, it is distributed into three main capsules. All three main capsules include statistical pooling which is mainly used for detecting the forgery image or videos. The outputs of three main capsules are dynamically routed to output capsules. Thus, the output capsules detect the image or video as real or fake. Algorithm 2 describes the capsule dual graph. The notations used in the Algorithm 2 are given in Table 2.

**Table 2   Notations used in Algorithm 2.**

| Notations | Description |
|---|---|
| $out_{j/i}$ | Output capsules generates in $j$, $i$ layer |
| $Wet$ | Weight tensor |
| $itr$ | Total number of iterations |
| squash | Non linear activation function |
| $inc_i$ | Input capsule in $i$ capsule layer |
| $O_j$ | Finaareoutput vector |
| $outs_j$ | Weight vector value for output capsules in $j$ layer |
| $out_{j/i}$ | Weighted prediction vector value for output Capsule in $j$, $i$ layer |
| $O_j$ | Vector of output capsule $j$ |

**Algorithm 2. Capsule Dual Graph.**

Step 1: Procedure Dynamic Routing $((out_{j/i},\ Wet, itr)$

Step 2: for $i$ in range do // DGN

Step 3: $\widehat{Wet}\ \leftarrow Wet\ +\ rand(size(Wet))$

Step 4: $\widehat{out}_j \leftarrow \widehat{Wet}_j \, \text{squash}(out_{j/i})$ where $Wet_i \in \mathbb{R}^{m \times n}$

Step 5: For all input $i$ capsules and output capsule $j$ do

Step 6: $bai_{ij} \leftarrow 0$

Step 7: End For

Step 8: For $itr$ iterations do

Step 9: For all input capsules $i$ do

Step 10: $inc_i \leftarrow \text{softmax}(bai_i)$

Step 11: For all output capsules $j$ do

Step 12: $outs_j \leftarrow \sum_i inc_{ij} \widehat{out}_{j/i}$

Step 13: For all output capsules $j$ do

Step 14: $o_j \leftarrow \text{squash}(sq_j)\ =\ \dfrac{\|sp_i\|^2}{1+\|sp_i\|^2} \cdot \dfrac{sp_i}{\|sp_i\|}$

Step 15: for all input $i$ capsules and output capsules $j$ do

Step 16: $bai_{ij} \leftarrow bai_{ij} + \widehat{out}_{j/i} \cdot o_j$

Step 17: End For

Step 18: return $o_j$

Step 19: End For

In Algorithm 2, the dynamic routing of three main output capsules $out_{j/i}$ for $it$ iterations are evaluated. To improve the efficiency of the algorithm, Gaussian noise is slightly added to the 3-D weight value of tensor $Wet$, and the squash is done in the following Eq. (10) before the process of routing through all iterations.

$$\text{squash}(x) = \frac{\|x\|^2}{1+\|x\|^2} \cdot \frac{x}{\|x\|}. \tag{10}$$

Thus, the added Gaussian noise helps to reduce the over-fitting of the graph. The outputs of the main capsules are calculated as:

$$L = -(x \log(\hat{x}) + (1-x) \log(1-\hat{x})) \tag{11}$$

where, $x$ is the ground truth value of the label and $\hat{x}$ is the predicted value of the label calculated using Eq. (9). The output capsule $o_j$ by using:

$$\hat{x} = \frac{1}{m} \sum_i \text{softmax} \left( \begin{bmatrix} o_1^T \\ o_2^T \end{bmatrix} \right). \tag{12}$$

By using Eq. (12), the length of the output capsules separates the two output capsules, for example, detection of real image and fake image or video for all dimensions.

## RESULT & DISCUSSIONS

### Data collection

This proposed work VJ-CN is implemented using the dataset CelebDF-FaceForencics++(c23) which is the combination of FaceForencies++ (c23) and Celeb-DF. For training the dataset, the dataset is randomly split into training subset data and a testing subset of data. Thus, the pixels of the image have normalized to the range of $(-1,1)$. FaceForencies++ dataset and Celeb-DF dataset are collected from *Rössler et al. (2019)*, *Yuezun Li & Lyu (2020)* and *Zhang et al. (2022a)*. DFFD and CASIA-WebFace data are collected from *Karras, Laine & Aila (2019)* and *Yi et al. (2014)*.

### Performance metric measures

The different metric measures are used in evaluating the performance of the proposed techniques. Some metrics such as accuracy calculation and error detection rate are probably used. To evaluate the proposed outcome and performance, comparative study on the existing models such as, To evaluate the proposed outcome and performance, the comparative study on existing models such as the VGG-19 (*Simonyan & Zisserman, 2014*), VIOLA-Jones (*Huang, Shang & Chen, 2019*), Capsule Net (*El Alaoui-Elfels & Gadi, 2021*), VIOLA-Jones +Capsule network is analyzed. Evaluation metrics measure the performance using accuracy, specificity, sensitivity, root mean square error (RMSE), mean absolute error (MAE), signal noise ratio (SNR), and peak signal noise ratio (PSNR). One of the evaluation metrics, an area under the receiver operating characteristic (AUROC) curve, is used to evaluate the real and fake images using the proposed method VJ-CN. The true positive ratio (TPR) is calculated using true positive (TP), true negative(TN), false positive (FP), and false negative (FN) values. Vice versa false positive ratio (FPR) is calculated as given in the below Eqs. (13)–(22) (*Jiang et al., 2022*; *Jiang & Li, 2022*; *Zhang et al., 2022b*).

$$TPR = \frac{TP}{TP+FN}, \tag{13}$$

$$FPR = \frac{FP}{FP+TN}, \tag{14}$$

$$Accuracy = \frac{TP + TN}{TP + TN + FP + FN} \cdot 100, \tag{15}$$

$$Recall = \frac{TP}{TP + FN}, \tag{16}$$

$$Precision = \frac{TP}{TP + FP}, \tag{17}$$

$$F1 - Measure = \frac{2 \cdot Precision \cdot Recall}{Precision + Recall}. \tag{18}$$

The error rate is given below:

$$PSNR = 20\log_{10}\left(\frac{255^2}{MAE}\right), \tag{19}$$

$$MAE = \frac{1}{MN}\sum_{i=1}^{M}\sum_{j=1}^{N}\left|X(i,j) - Y(i,j)\right|, \tag{20}$$

$$RMSE = \sqrt{\frac{1}{N}\sum_{i=1}^{N}\left(X_i - \widehat{X}_i\right)^2}, \tag{21}$$

$$SNR(db) = 20\log\left(\frac{V_{RMS(Signal)}}{V_{RMS(Noise)}}\right). \tag{22}$$

Table 3 presents the accuracy comparison on various datasets as shown.

Table 3 shows the accuracy measures of the proposed model, and it is used publicly with available datasets like FaceForencies++, Celeb-DF, DFFD, and CASIA-WebFace. The VJ-CN accuracy rate was high (94.92).

Table 4 shows the error rate of the above algorithms in the available data sets. Therefore, it is well understood that the proposed method VJ-CN has achieved a less error rate of 6.17 for the CASIA-WebFace data set. Figure 8 shows that the AUROC curve for the data sets Celeb-DF-FaceForensics++ (c23), DFFD, and the CASIA-WebFace data sets and VJ-CN has produced the best results in CASIA Web Face and Celeb-DF.

From the observation of Fig. 8, it is clearly shown that, for the dataset Celeb-DF-FaceForensics++(c23), the proposed VJ-CN method is produced with the best result compared to the other existing algorithms. Figure 9 shows the accuracy that is used to compute the error rate using Eqs. (20)–(22).

In Fig. 9, it is observed that the increase in PSNR and the MAE value is decreased for the best detection of fake faces in the video image and the faces of the real video image.

**Table 3   Performance comparison of proposed methods and accuracy on various datasets.**

| Methods | Datasets | | | |
|---|---|---|---|---|
| | FaceForensics++ (c23) | Celeb-DF | DFFD | CASIA-WebFace |
| VGG-19 | 84.51 | 73.12 | 86.33 | 88.25 |
| VIOLA-Jones | 85.16 | 77.12 | 87.45 | 86.23 |
| Capsule Graph Network | 87.67 | 79.12 | 87.78 | 89.45 |
| Proposed VJ-CN | 94.92 | 97.76 | 96.23 | 97.72 |

**Table 4   Comparison of error rate detection.**

| Methods | Datasets | | | |
|---|---|---|---|---|
| | FaceForensics++ (c23) | Celeb-DF | DFFD | CASIA-WebFace |
| VGG-19 (*Simonyan & Zisserman, 2014*) | 13.73 | 27.52 | 13.34 | 13.12 |
| VIOLA-Jones (*Huang, Shang & Chen, 2019*) | 14.57 | 24.12 | 14.85 | 14.88 |
| Capsule Graph Network (*El Alaoui-Elfels & Gadi, 2021*) | 12.31 | 15.45 | 14.21 | 12.24 |
| Proposed VJ-CN | 8.11 | 10.11 | 7.14 | 6.17 |

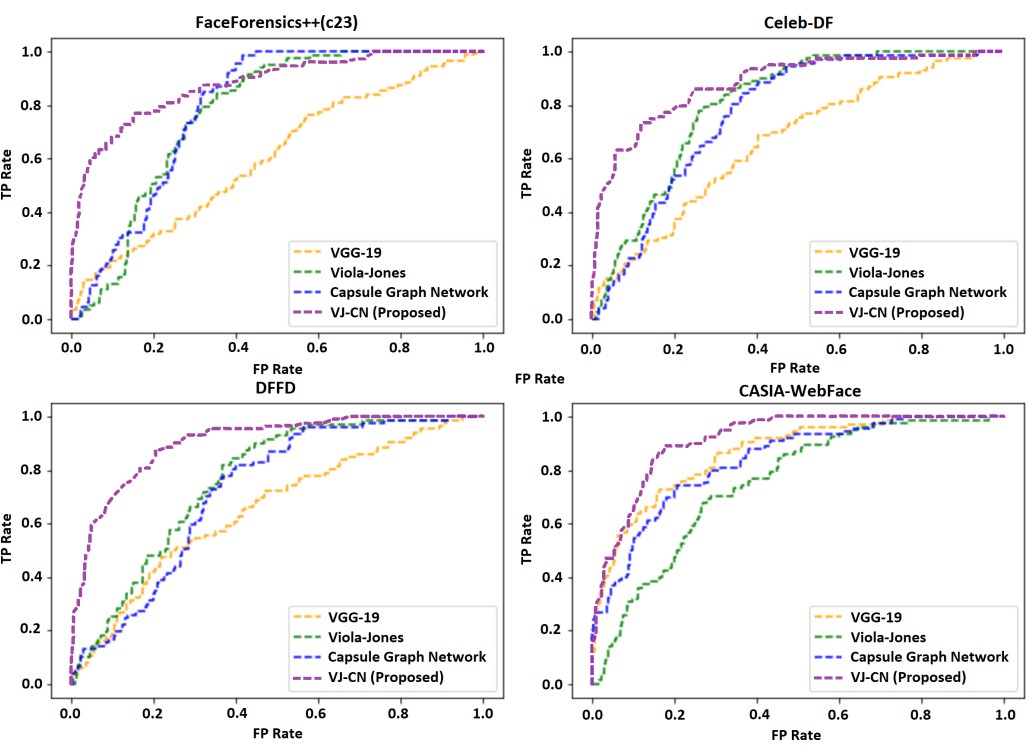

**Figure 8   Area under the receiver operating characteristic (AUROC) curve.**

The proposed work obtains a minimum error rate (*Kong et al., 2020*; *Wu, Jin & Yue, 2022*; *Wang, Han & Jin, 2022*). As a result, the value of PSNR is increased, and the MAE value is decreased compared to the existing models. Figure 9 represents the computation time

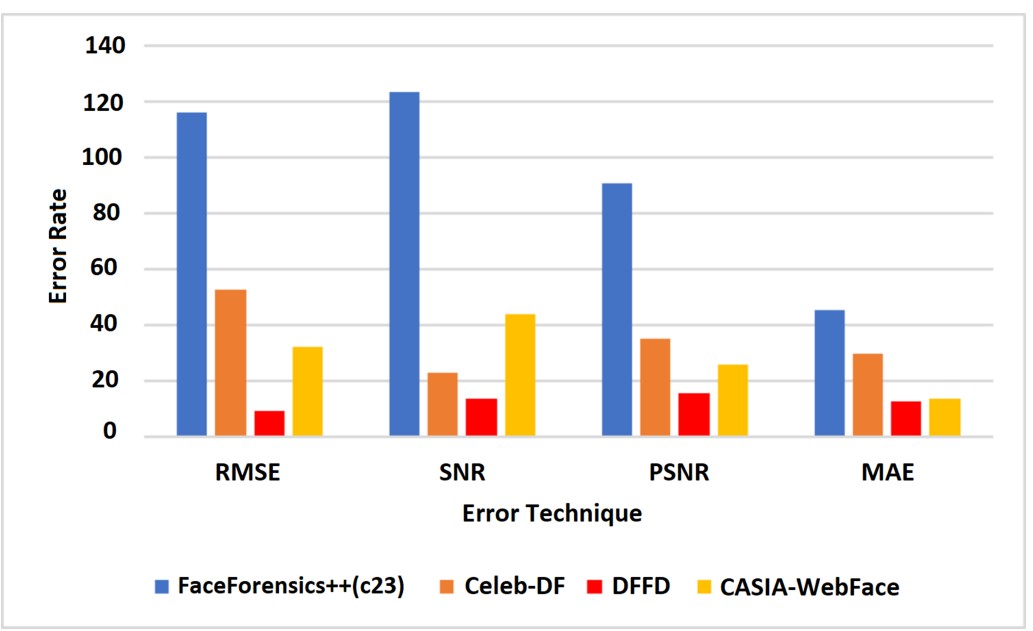

**Figure 9** Error rate in accuracy.

algorithms used in this research with various datasets like CelebDF-FaceForencics++ (c23), DFFD, and the CASIA-WebFace dataset.

Figure 10 shows that the proposed algorithm VJ-CN needs less time for computation when compared to the other existing techniques. It can produce input data validation regarding loss and accuracy for the forged image dataset of faces. The analysis of the training and testing performance of the dataset in the detection of fake face video images is provided. Figure 11 shows the precision of various data sets.

Figure 11 shows the precision metric measures of different data sets with different algorithms that are analyzed. Therefore, the proposed work provides the best performance compared to existing techniques and the above datasets such as Celeb-DF-FaceForensics++ (c23), DFFD, and CASIA-WebFace datasets. Here, the proposed VJ-CN method got a precision score equal to 85.88% in Celeb-DF-FaceForensics++ (c23) dataset, 88.91% in DFFD dataset, 90.35% in CASIA-WebFace dataset. Figure 12 shows the recall of various datasets.

Figure 12 shows recall metric measures of the different datasets with different algorithms that are analyzed. Therefore, the proposed work provides the best performance compared to the existing techniques and the above datasets of the CelebDF-FaceForencics++ (c23), DFFD, and CASIA-WebFace datasets. Here, VJ-CNof proposed work got a recall score of 91.42.

The proposed work provides the best performance compared to the existing techniques and the above datasets of CelebDF-FaceForencics++ (c23), DFFD, and the CASIA-WebFace

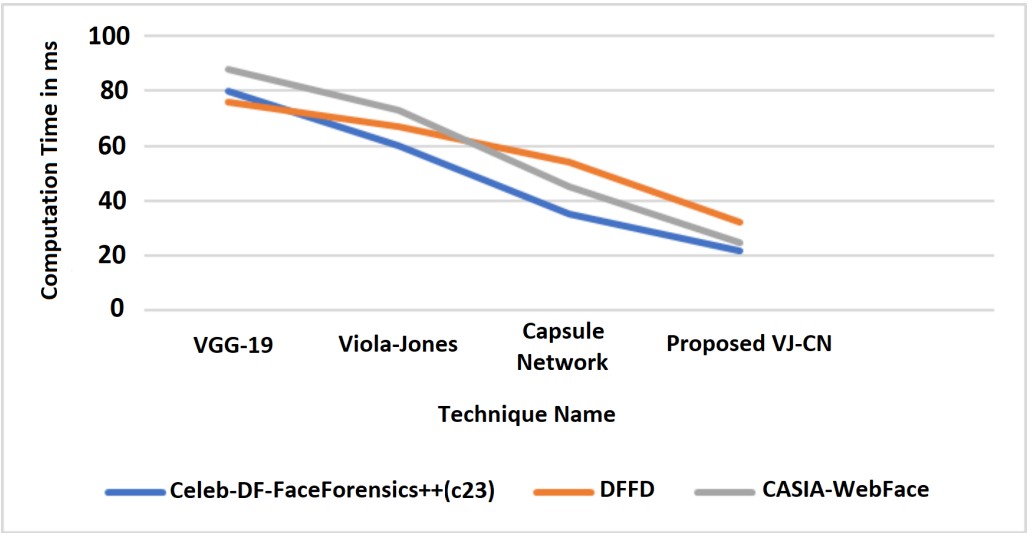

**Figure 10**  Computation time.

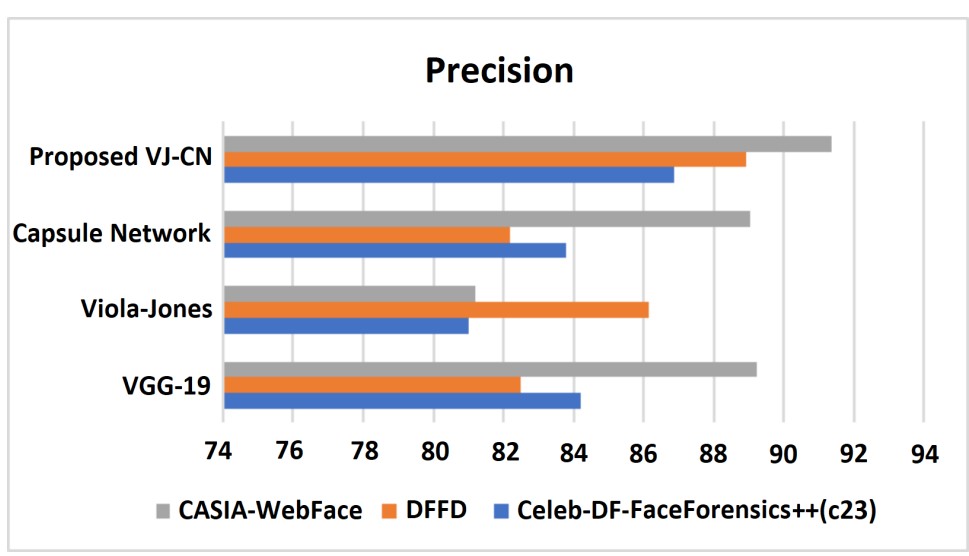

**Figure 11**  Precision metric measures of different data sets with different algorithms that are analyzed.

dataset. Table 5 shows the F1-Measure of the different datasets with different algorithms that are analyzed. Here, VJ-CNof proposed work got F1-Measure as 90.45.

## CONCLUSION

Face detection requires deep learning algorithms to fix the forged images very efficiently. Fake face detection is an NP-Hard problem that requires the intelligent model to identify the forged pixels. Fake faces can be presented with real images like formation across the internet. This makes to believe false assumptions and creates unnecessary social problems
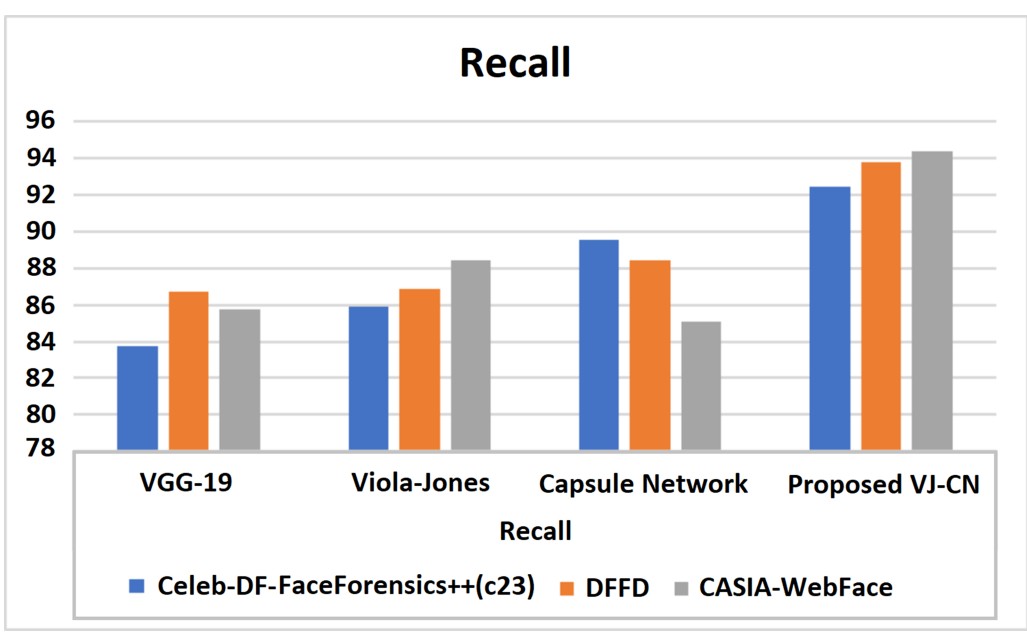

**Figure 12** Recall of various datasets.

**Table 5** *F*1— Measure of various datasets.

| Data sets | F1-Measure | | | |
|---|---|---|---|---|
| | VGG-19 *Simonyan & Zisserman (2014)* | VIOLA-Jones *Huang, Shang & Chen (2019)* | Capsule graph *El Alaoui-Elfels & Gadi (2021)* | VJ-CN (Proposed) |
| Celeb-DF-FaceForensics++(c23) | 81.34 | 83.12 | 88.89 | 90.45 |
| DFFD | 88.25 | 84.56 | 84.42 | 93.88 |
| CASIA-WebFace | 80.67 | 88.44 | 82.12 | 94.35 |

in society. This research implements the VIOLA-Jones algorithm to detect the faces from the input datasets. Then the predicted faces are analyzed using a capsule dual graph network. Capsule dual graph is a deep learning model that learns features' originality with high accuracy. The implementation uses a benchmark dataset related to fake faces. The features in the image are tracked as a node using the DGN algorithm, and unconnected nodes are identified as forged features. The result is evaluated using precision, recall, F1-Measure, error rate, and ROC/AUC for CelebDF-FaceForencics++ (c23) dataset, DFFD dataset, and CASIA-WebFace datasets. The error rate of the proposed algorithm is 8.11 for CelebDF-FaceForencics++ (c23), 7.14 for the DFFD dataset, and 6.17 for the CASIA-WebFace dataset. The prediction error is less than in existing models. However, our system has less error rate; there is a need for improvised in future work. The swarm-based optimization techniques are widely used to optimize the feature and improve accuracy.

Our future work may follow an optimization algorithm to improve the deep fake face detection rate.

### Funding

The research was supported by the Excellence Project Faculty of Science, University of Hradec Králové, No. 2210/2023-2024. The funders had no role in study design, data collection and analysis, decision to publish, or preparation of the manuscript.

### Grant Disclosures

The following grant information was disclosed by the authors:
Excellence Project Faculty of Science, University of Hradec Králová: 2210/2023-2024.

### Competing Interests

The authors declare there are no competing interests.

### Author Contributions

- Venkatachalam K conceived and designed the experiments, performed the computation work, prepared figures and/or tables, and approved the final draft.
- Pavel Trojovský conceived and designed the experiments, performed the experiments, analyzed the data, prepared figures and/or tables, authored or reviewed drafts of the article, and approved the final draft.
- Štěpán Hubálovský performed the experiments, analyzed the data, authored or reviewed drafts of the article, and approved the final draft.

### Data Availability

  The data is available at GitHub and Zenodo: https://github.com/venki202/VIOLA-Jones-algorithm;
  venkatachalam. (2023). VIOLA JONES ALGORITHM WITH CAPSULE GRAPH NETWORK FOR DEEPFAKE DETECTION [Data set]. Zenodo. https://doi.org/10.5281/zenodo.7743107.

### Supplemental Information

Supplemental information for this article can be found online at http://dx.doi.org/10.7717/peerj-cs.1313#supplemental-information.

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
