# Peer review of "VIOLA jones algorithm with capsule graph network for deepfake detection"

_PeerJ Computer Science, doi:10.7717/peerj-cs.1313_

## Round 0.1 · original submission · Major Revisions

· Academic Editor

Major Revisions

Based on reviewers comments, the authors are advised to make "major revisions".

Reviewer 1 ·

Basic reporting

The English language of the article is poor, needs rigorous revision on grammar and sentence formation.

The abstract is not highlighting the proposed technique clearly, rewrite it.

The reference formats inside the text are not correct, for example in the introduction section, there is no space in the author names, check it.

The introduction's last paragraph's first sentence is not correct, it should be four sections.

In the preprocessing step, the image is mentioned as forged - how is it defined here?

Give the reference for the Gabor filter.

Zooming augmentation is not defined clearly.

is it data set or dataset? - correct it.

Experimental design

In feature extraction, the second sentence is not correct - not a complete sentence, correct it.

In the Viola Jones algorithm - is it Haar Function or Haar Features? describe it clearly.

"The image is to be separated as 159 sub window and the features are related on random basis of the location" - this statement is not clear, correct it.

Provide the reference for the dataset used.

In the explanation of algorithm 2, the equation number is missing, provide it.

In page number 8/16, "The output of local consistency for hidden layer i is declared in Eqn (6)." is it equation 6 or 9 - check it.

In page number 8/16, line numbers 210, and 211 - the figure numbers are missing - provide it.

Validity of the findings

On page number 11/16 - line number 270 to 279, figure numbers are wrong and not specified. correct them.

Provide the explanation for PSNR - equation 18.

On page number 11/16, line number 264, equation numbers are mentioned as 20 to 23, but in the paper, only 21 equations are there - correct it.

Additional comments

The conclusion is convoluted, rewrite it clearly.

Reviewer 2 ·

Basic reporting

* grammar could be improved throughout
**** Line 59 contains a phrase that is not a full sentence: "So that computation time of the proposed work is reduced."
* citations sometimes are not properly spaced after text (e.g., in lines 34, 35, 37, 39).
Make sure that there is a space between the text (e.g., "AlexNet") and the citation (e.g., "Krizhevsky et al.")
* CNNs, RNNs, and GANs are mentioned in the literature section, but it does not review any transformer approaches.
Transformers have been used extensively in the past few years and have also been used for deepfake detection.
Some of this work should be mentioned as well.
* Line 91 contains an incorrect name. Line 91 should read "Generative Adversarial Network (GAN) forces the forensic"
not "Graph".
* There is some mention of GANs being used to generate synthetic images. StyleGAN should be included in this
discussion. The title is "A Style-Based Generator Architecture for Generative Adversarial Networks" with
authors Karras, Laine, and Aila.
* The formatting of text in Table 1 is not good. It is difficult to read, and spacing between words is missing or wrong.
* Table 1 is not referenced anywhere or explained. Its contents are not mentioned elsewhere in the paper, and
all of the entries need to be explained further.
* More explanation is needed on the shear mapping section. Please include equations and references.
* References for Haar functions should be included.
* References for AdaBoost should be included.
* Instead of using sudo code to in the algorithm boxes, use equations, text, and diagrams to describe the
algorithms.
* Lines 210 and 211 incorrectly reference figures and have "?"
* DGN is not officially defined. The first use of this acronym should be after the full phrase.
* Line 240 incorrenctly references an equation and has "?"
* Line 240 should use the name of this equation: binary cross entropy
* Acryonyms such as "TP", "FN", "TPR" are not defined in equations 12-21. Make sure to say the whole
name of these acronyms.

Experimental design

* The authors do an ablation study to determine how well their method compares to performance with
the independent components of VJ-CN. Results seem promising in that they are higher than the
components alone.
* More comparisons should be done. The authors mention many other
CNN and LSTM based methods in the Related Work section. Please include these results on these datasets
to show how proposed approach compares to deep learning methods already tested on these datasets.

Validity of the findings

* Figures 1 and 2 show data augmentations are done before data is split into training and testing sets.
This implies that data augmentations were applied to all of the data, including the testing data.
However, data augmentations should not be used on the testing data. They should only be applied to the training
data to increase the size of the training set, decrease overfitting, improve generalizability, etc.
The testing set should be the images/videos without augmentations so that the effectiveness of the method
can be determined on the true test set. This is especially important when comparing results on the same dataset
so that the comparison is valid.
* It is surprising that the authors chose K-means algorithm for feature extraction when there are a variety of
deep learning methods that can be used for feature extraction. The authors should elaborate on why they selected
this method compared to others.
* The ROC curves do not look correct. Is it AUC-ROC or ROC being reported? The maximum value of AUC or AUC-ROC is 1,
so none of the curves should exceed 1.
* The results section is disorganized and difficult to follow. Not all metrics are reported in tables, and not all metrics
are reported in graphs. It would be helpful to have them better organized and consistent to make comparisons.

---

## Round 0.2 · Minor Revisions

· Academic Editor

Minor Revisions

The authors are requested to make "minor revisions" and resubmit the manuscript.

Reviewer 1 ·

Basic reporting

Still English language of the article needs to improve.

The conclusion is not giving clear insight; rewrite it clearly by mentioning the problem, the solution proposed, and the performance evaluation of the solution;

Experimental design

satisfactory

Validity of the findings

satisfactory

---

## Round 0.3 · Minor Revisions

· Academic Editor

Minor Revisions

It seem that the comments from Reviewer 2 are not fully resolved. Kindly either resolve or respond to the comments.

Reviewer 1 ·

Basic reporting

This revision is satisfactory

Experimental design

This revision is satisfactory

Validity of the findings

This revision is satisfactory

Reviewer 2 ·

Basic reporting

The article reads better now.

Experimental design

The experiments still lack some completeness. I do not believe the authors have taken all my comments into consideration.

Validity of the findings

The findings are ok but still not impressive

Additional comments

I would like to see the authors respond to my review comments.

---

## Round 0.4 · accepted · Accept

· Academic Editor

Accept

The authors have resolved the issues of the reviewers and the paper is accepted.